# Profiling of In Vitro Bioaccessibility and Intestinal Uptake of Flavonoids after Consumption of Commonly Available Green Tea Types

**DOI:** 10.3390/molecules26061518

**Published:** 2021-03-10

**Authors:** Jeong-Ho Oh, Chan-Yang Lee, Yeong-Eun Lee, So-Hee Yoo, Jin-Oh Chung, Chan-Su Rha, Mi-Young Park, Yong-Deog Hong, Soon-Mi Shim

**Affiliations:** 1Department of Food Science and Biotechnology, Sejong University, 98 Gunja-dong, Seoul 143-747, Korea; jungho1223@naver.com (J.-H.O.); cks445@naver.com (C.-Y.L.); ungen0630@naver.com (Y.-E.L.); 0503ysh@naver.com (S.-H.Y.); 2AMOREPACIFIC R&D Center, 1920, Yonggu-daero, Giheung-gu, Yongin-si 17074, Gyeonggi-do, Korea; cjinoh@amorepacific.com (J.-O.C.); chansurha@gmail.com (C.-S.R.); mypark@amorepacific.com (M.-Y.P.)

**Keywords:** loose leaf tea, powdered tea, GTE, CATEPLUS™, bioaccessibility, intestinal uptake, epicatechins, flavonols

## Abstract

The aim of this study was to profile the bioaccessibility and intestinal absorption of epicatechins and flavonols in different forms of green tea and its formulation: loose leaf tea, powdered tea, 35% catechins containing GTE, and GTE formulated with green tea-derived polysaccharide and flavonols (CATEPLUS™). The bioaccessibillity and intestinal absorption of epicatechins and flavonols was investigated by using an in vitro digestion model system with Caco-2 cells. The bioaccessibility of total epicatechins in loose leaf tea, powdered tea, GTE, and CATEPLUS™ was 1.27%, 2.30%, 22.05%, and 18.72%, respectively, showing that GTE and CATEPLUS™ had significantly higher bioaccessibility than powdered tea and loose leaf tea. None of the flavonols were detected in powdered tea and loose leaf tea, but the bioaccessibility of the total flavonols in GTE and CATEPLUS™ was 85.74% and 66.98%, respectively. The highest intestinal absorption of epicatechins was found in CATEPLUS™ (171.39 ± 5.39 ng/mg protein) followed by GTE (57.38 ± 9.31), powdered tea (3.60 ± 0.67), and loose leaf tea (2.94 ± 1.03). The results from the study suggest that formulating green tea extracts rich in catechins with second components obtained from green tea processing could enhance the bioavailability of epicatechins.

## 1. Introduction

Tea, derived from *Camellia sinensis*, is the most widely consumed beverage in the world aside from water, and it exists in multiple varieties (black, oolong, green, and white), defined by its degree of oxidation prior to drying [1]. The health-promoting effects of green tea are mainly attributed to its polyphenol content, particularly epicatechins and flavonols [2,3]. The beneficial effects of catechins are attributed to their antioxidant, anticarcinogenic, antimicrobial, antiviral, anti-inflammatory, and antidiabetic properties [2,4,5]. Recently, it was found that a slight increase of total catechins and polyphenols was observed during the withering stage of Assam (*C. sinensis var. assamica*) green tea in the leaf processing step [6]. Beside the tea variety, brewing conditions, such as the infusion time and temperature, were found to make a significant difference in the content of polyphenols, in particular, catechins [1,7]. Green tea is commercially available in different forms, including loose leaf, bagged, and powdered tea, but recently, green tea dietary supplements, catechin-rich green tea extracts formulated with other ingredients, have gained consumers’ attention. In addition, it has been suggested that some specific types of green tea provide a different degree of potential health benefits compared to other green tea types [8]. The study found that powdered tea contained a higher amount of catechins, particularly EGCG, and showed higher antioxidant activities than loose leaf tea [2,9,10]. It was also suggested that green tea formulated with dietary chemicals could improve the bioavailability of catechins due to their interaction in the gastrointestinal tract [11,12,13,14]. In detail, green tea extract formulated with flavonol glycoside-rich fractions derived from green tea significantly increased the bioaccessibility of epicatechins from GTE, in particular EGCG and ECG [11,12,13,14]. The results from these studies imply that different forms and formulations could make a difference in the profiling of tea phenolic constituents and their metabolites as well as intestinal uptake, and strengthen the association between tea intake and specific health benefits. Although several studies have been conducted to measure the content of green tea polyphenols and their interaction with antioxidant activity according to different forms of green tea [15,16,17,18], there is limited research on profiling the flavonoids of green tea in commercially available forms other than the traditional preparation. It was recently found that free radical scavenging activities in green tea forms (loose leaf, bagged, and powdered tea), which are usually used for tea preparation at home, were not significantly different and suggested that tea forms would not generalize antioxidant activity [19]. These results imply that the pharmacological effect of green tea may be influenced by a different way, such as the bioaccessible and absorbed amounts of polyphenols. In other words, how many flavonoids are recovered during digestion and absorbed in the small intestine could be a critical factor in determining the health benefits from consuming different forms of green tea. Therefore, it is necessary to discover the post-consumption bioavailability of tea catechins of commercially available green tea and green tea extract supplements. The aim of this study was to profile flavonoids, such as epicatechins and flavonols, as well as to assess their bioaccessibility and intestinal uptake from commercially available forms of green tea (loose leaf, powdered tea, 35% catechins containing green tea extracts, and green tea extracts formulated with polysaccharides and flavonols obtained during green tea processing).

## 2. Results

### 2.1. Profiling of Epicatechins and Flavonols in Different Types of Green Tea Raw Materials

Table 1 shows the amount (mg/g of dry weight) of epicatechins and flavonols contained in various types of green tea as raw materials. The contents of EGC (27.24 ± 1.34 mg/g) and EGCG (64.07 ± 2.28 mg/g) in powdered tea, which comprised 81% total epicatechins (111.06 ± 2.72 mg/g). Similarly, it was 83% total epicatechins in loose leaf tea. The individual epicatechin contents in GTE (37.87 ± 1.55–154.54 ± 3.15 mg/g) were slightly small compared to those in CATEPLUS™ (38.05 ± 7.11–180.27 ± 29.18 mg/g). The content of EGC and EGCG in each sample constituted a larger proportion of total epicatechins than those of EC and ECG. The amount of individual catehcins in CATEPLUS™, which was formulated with secondary dietary components, such as polysaccharides and flavonols obtained during the processing of GTE, was higher than GTE. The content of total epicatechins in GTE and CATEPLUS™ was significantly about 2.8 times higher than those of powdered tea and loose leaf tea (*p* < 0.05). CATEPLUS™ showed the highest amount of total epicatechins (381.49 ± 58.00 mg/g of dry weight), followed by GTE, loose leaf tea, and powdered tea in decreasing order. The contents of targeted flavonols, including myricetin, quercetin, and kaempferol, were not detectable in powdered tea and loose leaf tea. However, GTE was found to have 4.49 ± 0.37 mg/g of myricetin while CATEPLUS™ contained 5.23 ± 0.23 mg/g of myricetin, 2.94 ± 0.05 mg/g of quercetin, and 1.97 ± 0.22 mg/g of kaempferol, resulting in 10.14 ± 0.47 mg/g of total flavonols.

### 2.2. Profiling of Epicatechins and Flavonols in Different Types of Green Tea According to Drinking Usage

Table 2 shows the amount of epicatechins and flavonols when powdered tea, loose leaf tea, GTE, and CATEPLUS™ were dissolved according to the drinking usage commercially available. According to drinking usage, the content of total epicatechins was 315.68 ± 6.96, 305.26 ± 0.78, 105.69 ± 2.26, 101.33 ± 1.98, 95.23 ± 7.99, and 22.56 ± 0.76 mg/g in Cateplus, GTE, powdered tea at 70 °C, powdered tea at 1 °C, powdered tea at 20 °C, and loose leaf tea, respectively. The amount of total epicatechins in powdered tea was not significantly different among the different temperatures of 1, 20, and 70 °C. Interestingly, the content of epicatechins decreased when loose leaf tea was dissolved in 70 °C water, compared to other tea products. CATEPLUS™ showed a higher amount of total epicatechins than powdered tea and loose leaf tea by 3 and 14 times, respectively. Meanwhile, the total epicatechin level in GTE was not significantly different from that in CATEPLUS™. In the case of flavonols, myricetin, quercetin, and kaempferol were below the limit of detection (LOD, 2 ppm at myricetin and quercetin, 1.25 ppm at kampferol) in the three powdered tea samples and loose leaf tea. Meanwhile, GTE contained 1.63 ± 0.08 mg/g of myricetin only. CATEPLUS™ contained the highest amount of total flavonols (3.77 ± 0.12 mg/g), comprising 2.02 ± 0.06 mg/g of myricetin, 1.18 ± 0.13 mg/g of quercetin, and 0.58 ± 0.02 mg/g of kaempferol.

### 2.3. Bioaccessibility and Intestinal Uptake of Epicatechins and Flavonols in Different Types of Green Tea

Epicatechins and flavonols in aqueous fraction from digesta after in vitro digestion of different green tea samples were measured (Table 3). The content of total epicatechins in powdered tea (2.43 ± 0.09 mg/g) was higher than that in loose leaf tea (0.29 ± 0.02 mg/g); however, it was not significantly different. Additionally, there was no significant difference between GTE and CATEPLUS™ in the content of total epicatechins, though GTE (67.31 ± 2.91 mg/g) showed a slightly higher amount of total epicatechins than CATEPLUS™ (59.03 ± 5.31 mg/g). In the case of total flavonols, GTE had a myricetin level of 1.4 ± 0.06 mg/g but CATEPLUS™ showed total flavonoids of 2.52 ± 0.07 mg/g, which was comprised of myricetin, quercetin, and kaempferol.

Figure 1 illustrates the bioaccessibility of epicatechins and flavonols by different drinking methods of each green tea after in vitro digestion. The bioaccessibility levels of total epicatechins of powdered tea, loose leaf tea, GTE, and CATEPLUS™ were 2.30 ± 0.13%, 1.27 ± 0.10%, 22.05 ± 1.06%, and 18.72 ± 2.01%, respectively, showing that GTE and CATEPLUS™ had significantly higher bioaccessibility than powdered tea and loose leaf tea by about 11 times (*p* < 0.05). The bioaccessibility of EC was the highest in GTE (78.15 ± 3.88%) and CATEPLUS™ (79.05 ± 2.12%), followed by ECG (53.05 ± 3.88% and 43.08 ± 3.65%), EGC (18.36 ± 2.30% and 15.64 ± 2.89%), and EGCG (8.21 ± 059% and 5.07 ± 1.33%). The bioaccessibility level of total flavonols in GTE (31.20 ± 1.30%) was higher than that in CATEPLUS™ (24.87 ± 0.66%), which shows the same trend with the bioaccessibility of myricetin. However, interestingly, the bioaccessibility levels of quercetin and kaempferol were determined at CATEPLUS™ only by 60% and 90%, respectively. Table 4 depicts the Caco-2 cellular uptake of epicatechins and flavonols from the aqueous fraction of green tea digesta. In the case of powdered tea, the amount of EGC and EGCG was 2.44 ± 0.68 and 1.16 ± 0.07 ng/mg protein, respectively, while EC and ECG were not detected. In loose leaf tea, the contents of EGC, EC, and EGCG were 1.88 ± 1.30, 0.38 ± 0.32, and 0.69 ± 0.03 ng/mg protein, respectively, but ECG was not detected. In the group of GTE, the amount of EGC was 43.85 ± 4.35 ng/mg protein, followed by EC (6.24 ± 4.15 ng/mg protein), EGCG (3.84 ± 1.00 ng/mg protein), and ECG (3.45 ± 1.25 ng/mg protein). In CATEPLUS™, the amount of each epicatechin was 151.90 ± 4.70 ng/mg protein for EGC, 9.09 ± 1.79 ng/mg protein for EC, 7.42 ± 1.20 ng/mg protein for EGCG, and 2.98 ± 2.20 ng/mg protein for ECG, respectively. The amount of total epicatechins was 3.60 ± 0.67 ng/mg protein in powdered tea, 2.94 ± 1.03 ng/mg protein in loose leaf tea, 57.38 ± 9.31 ng/mg protein in GTE, and 171.39 ± 5.39 ng/mg protein in CATEPLUS™. These results imply that the intestinal uptake of total epicatechins in GTE and CATEPLUS™ is higher than powdered tea and loose leaf tea by 18 and 53 times, respectively (*p* < 0.05). Meanwhile, the intestinal uptake of quercetin and kaempferol was measured in CATEPLUS™ at only 32.11 ± 1.89 and 34.16 ± 0.55 ng/mg protein, respectively.

## 3. Discussion

EGC and EGCG, which are main catechins of green tea, are regarded as the most important tea catechins because of their high contents in tea and health-promoting effects [20,21,22,23]. In the current study, EGC and EGCG constituted 78.0–82.5% of the total epicatechins from each green tea sample. Our results indicated that the total epicatechins level was slightly higher in loose leaf tea (135.85 ± 3.42 mg/g dry weight) than powdered tea (111.06 ± 2.72 mg/g dry weight), which is processed differently to fresh green tea leaf, but there was no significant difference (*p <* 0.05), indicating that the rolling processing applied to only loose leaf tea did not result in a significant difference in the content of total epicatechins (*p* < 0.05). However, a recent study reported that the content of EGCG and ECG was increased by roasting and drying, while it was decreased by the rolling step [6]. In the current study, a three-fold higher amount of epicatechins was detected in GTE (318.15 ± 12.95 mg/g dry weight) and CATEPLUS™ (381.49 ± 58.00 mg/g dry weight) compared to powdered tea and loose leaf tea. It implies that a proper extracting method, such as ethanolic extraction, and concentration could increase the total epicatechins, including EGC and EGCG.

Regarding drinking usage, selective extraction of EGC and EGCG-enriched fractions from green tea at different brewing temperature conditions and durations has also been explored using numerous methods [24]. In the current study, total epicatechins from each green tea sample prepared according to drinking usages were not that different from the total epicatechins of raw green tea materials (product itself) except for loose leaf tea (Table 2). In detail, loose leaf tea steeped in water at 70 °C for 1.5 min contained five-fold lower total epicatechins (22.56 mg/g dry weight) than powdered tea. The amount of catechins was different depending on the brewing temperature and time [25,26]. Similar to our finding, Fujioka et al. [27] reported that loose leaf tea had about a three-fold lower amount of catechins than powdered tea prepared in the traditional brewing manner (1 g/80 mL). It is plausible that the drinking usage for powdered tea could be preferred by consumers rather than loose leaf tea in terms of the contents of total epicatechins. In the case of GTE and CATEPLUS™, almost one-third of the flavonols were extracted based on the recommended consumption method, providing an extraction efficiency for green tea bioactive compounds. In detail, CATEPLUS™ (10.14 ± 0.47 mg/g dry weight) had two-fold greater flavonols than GTE (4.49 ± 0.37 mg/g dry weight) and myricetin was predominantly found. The results suggest that consuming GTE and CATEPLUS™ according to drinking usage could provide flavonols besides catechins.

Limited studies are available on the digestive recovery of catechins from green tea according to the type or consumption methods. The current study first estimated the bioaccessibility of epicatechins and flavonols from green tea and its extracts in commercially available forms (Table 3 and Figure 2). Our data revealed that EC (3.92–79.05%) and ECG (2.52–53.05%) were more stable than EGC (0.83–18.36%) and EGCG (0.79–8.21%) in all varieties. The absolute value of epicatechins in GTE was slightly higher than that in CATEPLUS™, while the flavonols were lower than in the CATEPLUS™. In a similar pattern to our findings, a previous study showed a similar tendency, elucidating that the bioaccessibilities of EC and ECG were higher than those of EGC and EGCG [12]. Another study also revealed that relatively lower amounts of EGCG (13.1%) and EGC (12.1%) than ECG (48.7%) and EC (69.2%) were found after the in vitro digestion [28]. These results imply that EGCG and ECG are preferred in the formation of semiquinone free radicals in the pyrogallol moiety of the B ring at near-neutral pH [29,30]. Most native catechins were not absorbed but were expected to be accumulated in the intestinal lumen. A previous study reported that the plasma antioxidant capacity was enhanced when green tea was consumed as a supplement in capsule form compared to green tea leaf [31]. For these reasons, several studies conducted research on enhancing the intestinal uptake of flavon-3-ols by second dietary components due to the modulation of their interactions [12,14,32]. For instance, the bioavailability of EGCG was enhanced by the co-administration of piperine due to the reduction of the small intestinal glucuronidation of EGCG [32]. It was recently demonstrated that the intestinal transport of major epicatechins by human intestinal Caco-2 cells was enhanced by adding quercetin, fisetin, or a mixture of quercetin and fisetin [13]. The current study confirmed that the total epicatechins absorbed by Caco-2 cells followed by digestion was significantly higher in CATEPLUS™ (171.39 ± 5.39 ng/mg protein), which was formulated with GTE with flavonol extracts and polysaccharides (GTE: 57.38 ± 9.31, powdered tea: 3.60 ± 0.67, loose leaf tea: 2.94 ± 1.03 ng/mg protein). This finding, in agreement with the result of a previous study [12], revealed that co-ingestion of GTE with flavonol-rich excipient foods increased the absorption of epicatechins by acting as an enhancer or modulator during digestion and intestinal absorption, respectively. In detail, catechol-containing flavonols serve as COMT inhibitors to be methylated instead of catechins or by modulating the gene expression of intestinal transporters. This could be a possible mode of action for enhancing the cellular uptake of epicatechins by utilizing flavonol-rich foods that can easily be incorporated into a daily diet.

## 4. Materials and Methods

### 4.1. Chemicals and Reagents

Epigallocatechin (EGC), epigallocatechin-3-gallate (EGCG), epicatechin gallate (ECG), and epicatechin (EC) were obtained from Wako (Osaka, Japan). Monopotassium phosphate (KH_2_PO_4_), dipotassium phosphate (K_2_HPO_4_), α-amylase, pepsin, lipase, bile acid, pancreatin, sodium bicarbonate (NaHCO_3_), fisetin, myricetin, kaempferol, quercetin, and formic acid were purchased from Sigma Aldrich (St. Louis, MO, USA). Sodium hydroxide and hydrogen chloride (HCl) were purchased from Daejung chemicals and metals (Gyeonggi-do, South Korea). Triple distilled water was purchased from Ultra 370 series, Gyeonggi-do, South Korea. Phosphoric acid was purchased from Junsei Chemical (Tokyo, Japan). Acetonitrile, water, methanol, and HPLC-grade distilled water were purchased from J.T.Baker (Phillipsburg, NJ, USA). Dimethyl sulfoxide (DMSO) with 50% MeOH was purchased from Duksan (Ansan, Gyeonggi-do, South Korea).

### 4.2. Sample Preparations

Differently processed green tea, such as loose leaf tea, powdered tea, GTE, and CATEPLUS™, were obtained from the AMOREPACIFIC R&D Center, and the overall preparation scheme is shown in Figure 2. Further details regarding the processing methods for tea preparation were described in a previous study [33]. General drinking methods of green tea were entirely different from 1.5 g/200 mL to 2 g/100 mL [3,11,16,19,34]. In order to evaluate the amount of green tea flavonoids according to the brewing method, a general drinking method from AMOREPECIFIC, which is a popular tea company in Korea, was applied to each type of green tea: powdered tea was suspended in 1, 2, and 70 °C water in proportions of 0.15 g/100 mL; loose leaf tea was soaked for 1.5 min in 70 °C water at a concentration of 1.5 g/300 mL of water; GTE was suspended in 1 g/115 mL of water at 20 °C; and CATEPLUS™ was suspended in 1.2 g/ 115 mL of water at 20 °C. The reason that powdered tea was treated at only 70 °C was that the content of epicatechin at 1, 20, and 70 °C was not significantly different when digested according to the drinking methods (Table 2). Because of their high average content, the samples treated at 70 °C were used in the experiment. Then, 20mM phosphate buffer (PB) was prepared by dissolving potassium biphosphate and dipotassium phosphate in water.

### 4.3. Measurements of the Bioaccessibility of Green Tea Flavonoids by Using an In Vitro Digestion Model System

The in vitro digestion model system simulating the human gastrointestinal tract with salivary, gastric, and small intestinal phases was adopted from a previous study [35,36]. All digestive enzymes were kept in ice at 4 °C. Then, 2 mL of amylase (one unit from human saliva per mL 20 mM PB) were added, and the mixture was incubated in a water bath at 37 °C, 200 rpm for 3–5 min during the salivary phase. The initial pH in the gastric phase was regulated to 2.0 ± 0.1 via the addition of 1 M HCl, and 4 mL of pepsin (40 mg per mL 0.1M HCL) were added to the solution. After incubation in trembling water for 1 h (37 °C, 200 rpm), 1 M NaHCO3 was added to adjust the pH 5.3 ± 0.1 to mimic the conditions in the upper intestine, followed by the addition of a solution of small intestinal enzymes (2 mL pancreatin (2.0 mg/mL 20 mM PB), 2 mL porcine lipase (1.0 mg/mL 20 mM PB), and 2 mL bile acid (12.0 mg/mL 20 mM PB)). Then, 1.0 M NaOH was used to adjust pH 7.0 ± 0.1 to mimic the conditions in the small intestine, and the solution was placed in a shaking water bath for 2 h at 37 °C, 200 rpm. The final volume of all samples was adjusted to 17.5 mL. After in vitro digestion, the samples were centrifuged at 10,000 rpm (4 °C, 5 min) and the supernatant was filtrated using a syringe filter with a pore size of 0.2 µm (Advantech, Tokyo, Japan).

Bioaccessibility (%), the relative amount transferred from raw materials to aqueous fraction from digesta after in vitro digestion, was calculated through the following equation:(1)$$=\frac{\;C_{d}\left(\frac{\mu g}{\mathrm{mL}}\right)\times final\;volume\;\left(17.5\;\mathrm{mL}\right)\times Dilution\;factor\;}{Concentration\;of\;epicatechins\;or\;flavonols\;from\;raw\;material\;\left(\mathrm{mg}/g\right)}$$
where *C_d_* is the content of epicatechins or flavonols in aqueous fraction from digesta, and the dilution factor means the final volume for in vitro digestion.

### 4.4. Measurements of Intestinal Uptake of Green Tea Flavonoids by Caco-2 Cell Culture

The intestinal cellular uptake of green tea catechins was conducted using Caco-2 cells (KCLB, Seoul, South Korea) between passages number 50 and 55. Caco-2 cells were used for experiments 14–21 days post-seeding onto a 90-mm plate (SPL Life Science, Gyeonggi-do, South Korea). The medium was altered daily, and the plate was washed with a phosphate buffer solution (PBS, Corning). The epithelial cells were seeded onto a 12-well plate (Corning) at 1 × 10^5^ cells per well, grown in Dulbecco’s modified Eagle’s medium (DMEM; Corning) with 10% fetal bovine serum (FBS; Biopure, Cambridge, MA, USA) and 1% penicillin/streptomycin (Biotechnics Research, Inc, Lake Forest, CA, USA), and incubated at 37 °C under an incubator atmosphere of air/CO_2_ (95:5) until 100% confluency was achieved. Before treatment, a phosphate buffer solution (PBS, Corning) was used to wash the cells to starve the cells of FBS for 30 min in 5% CO_2_ to promote incubation. An aliquot amount of (0.5 mL) aqueous fraction was dispersed into each well after mixing DMEM (1:1, *V:V*). After incubation in 5 % CO_2_ for 2 h, the cell medium in the 12-well plate was removed, and the cells were washed with PBS. Protease (0.01 g/mL PBS) was added to each well to detach the Caco-2 cells, and the cells were removed by pipette suction. The obtained cells were centrifuged and sonicated in a water bath for 30 min. The supernatant liquid was obtained and stored at −20 °C after the pH was adjusted to 3.0 ± 0.1 via the addition of 0.2% phosphoric acid (pH 1.9). Cellular differentiation and polarity were checked by morphology. The intestinal uptake of epicatechins and flavonols was calculated by measuring their amount in cells and then adjusted by cellular protein (mg of protein).

### 4.5. Simultaneous Analysis of Green Tea Flavonoids (Catechins and Flavonols) Using UPLC-PDA-ESI/MS

Four major catechins (EC, EGC, ECG, and EGCG) and flavonols (myricetin, quercetin, and kaempferol) were simultaneously identified and quantified by using ultra-performance liquid chromatography (UPLC, Thermo fisher scientific) with a photodiode array (PDA) detector/electrospray ionization/mass spectrometry (ESI/MS). Before analysis, each sample was mixed with 50% methanol and 0.2% phosphoric acid. The mixtures were filtered through PTFE syringe filters (0.45 μm, ADVENTEC) and dissolved in 50% methanol. A Poroshell 120 C18-column (4.6mm × 150mm, 2.7 μm, Agilent) was used to separate the compounds at 280 nm. The composition of the mobile phase was water containing 0.1% formic acid (A) and acetonitrile containing 0.1% formic acid (B). The protocols of the gradient elution were as follows: 8% B (0–2 min), 12% B (3 min), 16% B (4–12 min), 20% B (15–18 min), 24% B (21 min), 30% B (22–26 min), 50% B (28–30 min), 80% B (32 min), and 8% B (34–35 min). The flow rate was 0.8 mL/min with an injection volume of 5 μL, and the column temperature was set at 30 °C. MS scanning was performed in a negative mode in the range of *m/z* 100 to 600. The conditions of ESI/MS were used at 35 V of capillary voltage, 270 °C capillary temperature, 40 arb of auxiliary gas flow, and 75 arb of sheath gas flow.

### 4.6. Statistical Analysis

The values are reported as the mean ± standard deviation (SD). One-way analysis of variance (ANOVA) and Tukey’s post hoc test were performed to measure significant differences among the groups at a significance level of *p* < 0.05 using Graphpad Prism 3.0 software (Graphpad, CA, USA). All real-time experiments were performed in triplicate from two independent experiments, and the statistical analysis was determined using a Student’s t test.

## 5. Conclusions

In conclusion, we presented the first profiling of epicatechins and flavonols according to the consumption methods and their bioaccessibility and intestinal uptake in various commercially available green tea products, which were differently processed. A significant amount of total epicatechins and their bioaccessibility was found in both GTE and CATEPLUS™ compared to loose leaf tea and powdered tea. There was no significant difference in the bioaccessibility of total epicatechins between CATEPLUS™ and GTE, but the intestinal uptake of total epicatechins was three times higher in CATEPLUS™ than GTE. CATEPLUS™ had more bioaccessible flavonols, and taken up well by the intestinal cell. The results from this study suggest that GTE formulated with second dietary components obtained from green tea processing (i.e., polysaccharide and flavonol extracts) could enhance the bioavailability of epicatechins as well as flavonols. It also suggests that specific bioactivity could be diverse in differently processed teas and forms. Therefore, it is necessary to discover the post-consumption bioavailability of tea catechins of commercially available green tea and green tea extract supplements. Further elucidation of metabolic processes, such as glucuronization, sulfation, and methylation after the digestive process of catechin, is required in future studies.

## Figures and Tables

**Figure 1 molecules-26-01518-f001:**
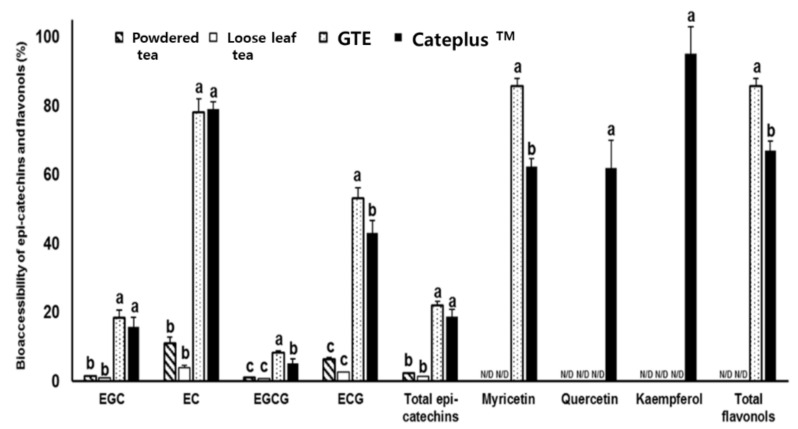
Bioaccessibility of epicatechins and flavonols (%). GTE: 35% catechins containing extract, CATEPLUS™: GTE formulated with green tea-derived polysaccharide and flavonols. Different letters on each bar indicate a significant difference within groups (*p* < 0.05). N/D: not detected below 1 μg/mL (EGC, EC, EGCG, ECG) or 1.25 μg/mL (Myricetin, Quercetin, Kaempferol).

**Figure 2 molecules-26-01518-f002:**
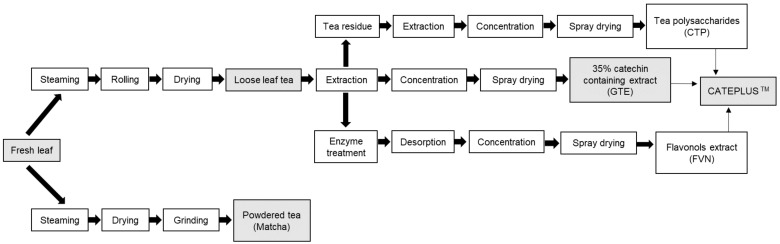
Scheme for each processing condition in green tea leaves (loose leaf tea and powdered tea) and its commercialized extracts (35% catechins containing green tea extracts, GTE formulated with polysaccharides and flavonols). GTE: 35% catechins containing extract.

**Table 1 molecules-26-01518-t001:** Content of epicatechins and flavonols in commercially available raw green tea materials (product itself) (mg/g dry weight).

Types of Green Tea	Sample Preparation	EGC	EC	EGCG	ECG	Total Epicatechins	Myricetin	Quercetin	Kaempferol	Total Flavonols
Powdered tea	Raw material (in 20% DMSO with 50% MeOH)	27.24 ± 1.34 ^b^	6.35 ± 0.73 ^c^	64.07 ± 2.28 ^b^	13.4 ± 0.68 ^b^	111.06 ± 2.72 ^b^	< LOD *	< LOD	< LOD	< LOD
Loose leaf tea	Raw material (in 20% DMSO with 50% MeOH)	37.25 ± 1.49 ^b^	9.49 ± 0.64 ^c^	74.87 ± 0.32 ^b^	14.23 ± 1.49 ^b^	135.85 ± 3.42 ^b^	< LOD	< LOD	< LOD	< LOD
GTE	Raw material (in 20% DMSO with 50% MeOH)	98.46 ± 8.27 ^a^	27.29 ± 0.99 ^b^	154.54 ± 3.15 ^a^	37.87 ± 1.55 ^a^	318.15 ± 12.95 ^a^	4.49 ± 0.37 ^b^	< LOD	< LOD	4.49 ± 0.37 ^b^
CATEPLUS™	Raw material (in 20% DMSO with 50% MeOH)	115.49 ± 14.00 ^a^	38.05 ± 7.11 ^a^	182.27 ± 29.18 ^a^	45.68 ± 8.85 ^a^	381.49 ± 58.00 ^a^	5.23 ± 0.23 ^a^	2.94 ± 0.05	1.97 ± 0.22	10.14 ± 0.47 ^a^

* LOD: Limit of detection; EGC, EC, EGCG, ECG: 2 ppm; myricetin, quercetin: 2 ppm; kaempferol; 1.25 ppm. GTE: 35% catechins containing extract, CATEPLUS™: GTE formulated with green tea derived polysaccharide and flavonols, EGC: epigallocatechin, EC: epicatechin, EGCG: epigallocatechin gallate, ECG: epicatechin gallate. Different letters (a–c) indicate significant difference within groups (*p* < 0.05).

**Table 2 molecules-26-01518-t002:** Epicatechins and flavonols from green tea samples according to drinking usage (mg/g dry weight).

Types of Green Tea	Drinking Usage	EGC	EC	EGCG	ECG	Total Epicatechins	Myricetin	Quercetin	Kaempferol	Total Flavonols
Powdered tea	0.15 g/100 mL H_2_O @1 °C	24.08 ± 1.03 ^b^	5.95 ± 0.54 ^c^	60.52 ± 0.83 ^b^	10.78 ± 0.52 ^b^	101.33 ± 1.98 ^b^	< LOD *	< LOD	< LOD	< LOD
	0.15 g/100 mL H_2_O @20 °C	23.78 ± 1.50 ^b^	5.45 ± 0.15 ^c^	56.77 ± 4.69 ^b^	9.23 ± 1.69 ^b^	95.23 ± 7.99 ^b^	< LOD	< LOD	< LOD	< LOD
	0.15 g/100 mL H_2_O @70 °C	25.77 ± 0.88 ^b^	7.01 ± 0.95 ^c^	62.49 ± 1.20 ^b^	10.43 ± 0.56 ^b^	105.69 ± 2.26 ^b^	< LOD	< LOD	< LOD	< LOD
Loose leaf tea	1.5 g/300 mL H_2_O@70 °C, 1.5 min	8.81 ± 0.67 ^c^	2.31 ± 0.20 ^d^	9.49 ± 0.14 ^c^	1.96 ± 0.15 ^c^	22.56 ± 0.76 ^c^	< LOD	< LOD	< LOD	< LOD
GTE	1 g/115 mL H_2_O @20 °C	89.24 ± 6.21 ^a^	25.54 ± 0.26 ^b^	156.14 ± 2.41 ^a^	34.36 ± 0.78 ^a^	305.26 ± 0.78 ^a^	1.63 ± 0.08^b^	< LOD	< LOD	1.63 ± 0.08 ^b^
CATEPLUS™	1.2 g/115 mL H_2_O @20 °C	90.39 ± 6.82 ^a^	27.60 ± 0.30 ^a^	163.05 ± 377 ^a^	34.64 ± 0.59 ^a^	315.68 ± 6.96 ^a^	2.02 ± 0.06 ^a^	1.18 ± 0.13	0.58 ± 0.02	3.77 ± 0.12 ^a^

* LOD: Limit of detection; EGC, EC, EGCG, ECG: 2 ppm; myricetin, quercetin: 2 ppm; kaempferol; 1.25 ppm. GTE: 35% catechins containing extract, CATEPLUS™: GTE formulated with green tea derived polysaccharide and flavonols, EGC: epigallocatechin, EC: epicatechin, EGCG: epigallocatechin gallate, ECG: epicatechin gallate. Different letters (a–c) indicate significant difference within groups (*p* < 0.05).

**Table 3 molecules-26-01518-t003:** Contents of epicatechins and flavonols in aqueous fraction from digesta after in vitro digestion (mg/g dry weight).

Types of Green Tea	Drinking Usage	EGC	EC	EGCG	ECG	Total Epicatechins	Myricetin	Quercetin	Kaempferol	Total Flavonols
Powdered tea	0.15 g/100 mL H_2_O @70 °C	0.37 ± 0.12 ^b^	0.75 ± 0.04 ^c^	0.64 ± 0.07 ^b^	0.67 ± 0.08 ^b^	2.43 ± 0.09 ^b^	< LOD *	< LOD	< LOD	< LOD
Loose leaf tea	1.5 g/300 mL H_2_O @70 °C, 1.5 min	0.07 ± 0.01 ^b^	0.09 ± 0.01 ^c^	0.07 ± 0.00 ^b^	0.05 ± 0.01 ^b^	0.29 ± 0.02 ^b^	< LOD	< LOD	< LOD	< LOD
GTE	1 g/115 mL, H_2_O @20 °C	16.32 ± 1.47 ^a^	19.96 ± 1.02 ^b^	12.82 ± 1.04 ^a^	18.21± 0.72 ^a^	67.31 ± 2.91 ^a^	1.4 ± 0.06 ^a^	< LOD	< LOD	1.4 ± 0.06 ^b^
CATEPLUS™	1.2 g/115 mL, H_2_O @20 °C	14.07 ± 2.17 ^a^	21.81 ± 0.49 ^a^	8.24 ± 2.02 ^a^	14.91 ± 1.06 ^a^	59.03 ± 5.31 ^a^	1.25 ± 0.01 ^a^	0.72 ± 0.04	0.55 ± 0.03	2.52 ± 0.07 ^a^

* LOD: Limit of detection; EGC, EC, EGCG, ECG: 2 ppm; myricetin, quercetin: 2 ppm; kaempferol; 1.25 ppm. GTE: 35% catechins containing extract, CATEPLUS™: GTE formulated with green tea derived polysaccharide and flavonols, EGC: epigallocatechin, EC: epicatechin, EGCG: epigallocatechin gallate, ECG: epicatechin gallate. Different letters (a–c) indicate significant difference within groups (*p* < 0.05).

**Table 4 molecules-26-01518-t004:** Content of epicatechins and flavonols in Caco-2 cells (ng/mg protein).

Types of Green Tea	Drinking Usage	EGC	EC	EGCG	ECG	Total Epicatechins	Myricetin	Quercetin	Kaempferol	Total Flavonols
Powdered tea	0.15 g/100 mL H_2_O @70 °C	2.44 ± 0.68 ^c^	N/D	N/D	N/D	3.60 ± 0.67 ^c^	N/D	N/D	N/D	N/D
Loose leaf tea	1.5 g/300 mL H_2_O @70 °C, 1.5 min	1.88 ± 1.30 ^c^	0.38 ± 0.3 ^b^	0.69 ± 0.03 ^c^	N/D	2.94 ± 1.03 ^c^	N/D	N/D	N/D	N/D
GTE	1 g/115 mL H_2_O @20 °C	43.85 ± 4.35 ^b^	6.24 ± 4.15 ^ab^	3.84 ± 1.00 ^b^	3.45 ± 1.25	57.38 ± 9.31 ^b^	N/D	N/D	N/D	N/D
CATEPLUS™	1.2 g/115 mL H_2_O @20 °C	151.90 ± 4.70 ^a^	9.09 ± 1.79 ^a^	7.42 ± 1.20 ^a^	2.98 ± 2.20	171.39 ± 5.39 ^a^	N/D	32.11 ± 1.89	34.16 ± 0.55	66.27 ± 1.76

EGC, EC, EGCG, ECG: 2 ppm; myricetin, quercetin: 2 ppm; kaempferol; 1.25 ppm. GTE: 35% catechins containing extract, CATEPLUS™: GTE formulated with green tea derived polysaccharide and flavonols, EGC: epigallocatechin, EC: epicatechin, EGCG: epigallocatechin gallate, ECG: epicatechin gallate. Different letters (a–c) indicate significant difference within groups (*p* < 0.05). N/D: not detected below 1 μg/mL (EGC, EC, EGCG, ECG) or 1.25 μg/mL (Myricetin, Quercetin, Kaempferol).

## Data Availability

The data presented in this study are available on request from the corresponding author.

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
