# Peer review of "Profiling of In Vitro Bioaccessibility and Intestinal Uptake of Flavonoids after Consumption of Commonly Available Green Tea Types"

_molecules, 2021, doi:10.3390/molecules26061518_

Round 1

Reviewer 1 Report

The manuscript, "Profiling of bioaccessibility and intestinal uptake of flavonoids after consumption of commonly available green tea types", sent for review has been significantly improved compared to the previous version. It is much more readable, but still needs some corrections to be published.

Detailed comments:

line 35 - C. sinensis should be in italics

line 73-80 - all results are still listed, repeating with the table, please describe the most important, or specify a range.

chapter 2.2 - also the results are repeated with the table, please discuss the most important, significant, not all.

Unfortunately, the use of different dosages and volumes for tea extraction is still unclear 

Reviewer 2 Report

Oh et al. assessed in vitro bioaccessibility and intestinal uptake through Caco2-cell model epithelia of polyphenols from different green tea preparations. Although the experimental workflow appears well conceived, the originality of the research is limited. The authors take for granted that in vitro models utilized have any physiological relevance. This is not true in general and in vivo fate of phytochemicals of tea might greatly differ. In my opinion, the papers should include a specific section indicating the limits of the method, emphasizing why it is expected that polyphenols may change in (simulated) gastrointestinal tract and which enzymes or digestive components is any might change the chemical nature of tea polyphenols.

The authors carried out a HPLC and MS analysis of tea polyphenols; it would be interesting to check if neoformed compounds appear after simulated digestion, due to products of polyphenol metabolism. To this purpose, I would be curious of seeing some chromatogram after in vitro digestion and after Caco-2 monolayer uptake, for instance as Supplementary data. Finally, title and abstract should clearly indicated that bioaccessibility and uptake are studied in vitro.

The abstract does not contain any word about the method used, such as simulated digestion and Caco-2 cellular uptake. This information is mandatory, otherwise writing about "intestinal absorption" is misleading. Abbreviations must be defined at first usage.

L. 315. check the exponent of cells x well.

Round 2

Reviewer 1 Report

The paper has been improved, no further comments.

Reviewer 2 Report

The main concerns raised with the first report have been convincingly addressed. In my opinion the paper can be accepted in its present form.

This manuscript is a resubmission of an earlier submission. The following is a list of the peer review reports and author responses from that submission.

Round 1

Reviewer 1 Report

The manuscript "Profiling of bioaccessibility and intestinal uptake of flavonoids after consumption of commonly available green tea types" presented for review is very interesting and innovative. The methods used are correct. The manuscript needs to be refined in order to be published. Green tea bioaccessibility is a very important topic in human nutrition, hence publications on this topic are very important for the development of science.

Detailed comments:

Tables and figures should not be in a separate chapter, but under the paragraph discussing the data.

The data is described in great detail and repeated with the data in the tables, it is worth describing only the most important ones that have the most important value.

p in statistical descriptions (p <0.05) should be in italics.

Figure 2 - please describe the abbreviations N / A, how the statistical analysis was carried out (what the letters a-c mean)

Table 1 - could not the same dose and volume be used? The results would be easier to compare. There is no statistical description in the description of the table.

Table 1 and Table 2 show the same results.

Table 3 and 4 - no description of the statistical method (explanations of letters a and b)

line 196 - no reference [8] in the citation of Weiss and Anderton (2003)

The discussion of the results is not very extensive, it is worth adding a few more articles to it.

line 267 - 1.5min (no spaces)

4.5 and 4.6 - the same chapter title

Reviewer 2 Report

Profiling the bioaccessibility after consumption of green tea is interesting. However, I don't think that the setting conditions for the experiments were appropriate.

How did the author decide the drinking usage? The authors used 300 ml of water for 1.5 g of loose tea, so the amount of eluted catechins and other components was less than when drinking in general.

The most important issue is matcha. The authors used only 0.15 g of matcha for 100 mL of water. Why did the authors use such a small amount? Matcha is taken by suspending 2 to 3 g in 50 to 100 mL water, so the experimental dose was 1/10 to 1/20 of the general intake.

These inappropriate experimental conditions can lead the authors to a false evaluation.

Table 1 and Table 2 are the same.

The data of catechin etc. can be found by looking at the table, so all each value need not write again in the results.

In fact, how many mL of green tea samples was used in the in vitro digestion model system test?

The effects of digestive enzymes are considered in this experimental system, but what do the authors think about the involvement of intestinal bacteria?